# Identification of Fungal Communities Isolated from Himalayan Glacier Cryoconites

**Gandhali M. Dhume [1], Masaharu Tsuji [2] and Shiv Mohan Singh [3,4,5,*]**

1 Department of Microbiology, Goa University, Taleigao 403206, India
2 National Institute of Technology, Asahikawa College, Hokkaido 071-8142, Japan
3 National Centre for Polar and Ocean Research, Vasco-da-Gama 403804, India
4 Department of Botany, Institute of Science, Banaras Hindu University, Varanasi 221005, India
5 Science and Engineering Research Board, Department of Science and Technology, New Delhi 110016, India
* Correspondence: drshivmohansingh@gmail.com; Tel.: +91-942-076-7319

**Abstract:** The current study focuses on fungi that were isolated from cryoconite holes of the Hamtah glacier in the Himalayas. Cryoconite holes have ecological and biotechnological importance. A total of seven cryoconite samples were collected from different locations and subjected to the isolation of psychotropic fungi at 1, 4, 15 and 22 °C. Isolates were identified by ITS and D1/D2 region sequences. The result showed culturable yeasts (45) and filamentous fungi (10) belonging to four ascomycetous classes (Dothideomycetes, Eurotiomycetes, Saccharomycetes and Sordariomycetes) and two basidiomycetes' classes (Microbotryomycetes and Tremellomycetes). Physiological characteristics such as the pH, temperature, salt tolerance, carbon source utilization and antibiotics sensitivity of the isolates were studied. All the isolates were grown from acidic to alkaline pH and were able to grow at 1 to 22 °C. The fungal cultures isolated were screened to produce cold active enzymes such as amylase, cellulase, lipase, protease and catalase. Cellulase activity was detected at its maximum at both 4 and 15 °C. Himalayan cryoconites fungi showed immense potential for biotechnological and industrial applications. To the best of our knowledge, this is the first record of the characterization of fungal communities present in the glacier cryoconites of the Himalayas.

**Keywords:** cryoconites; diversity; fungi; biopotential; Himalaya

## 1. Introduction

Large parts of terrestrial and aquatic ecosystems of our planet seasonally or permanently show cold temperatures. Polar regions make up about 14% of the Earth's surface [1]. The Himalayas are a unique place in the mountain ecosystems of the world. They spread over 2500 km, starting from west–northwest to the east. The Himalayan region has major concentrations of glaciers and is covered by snow during winter., therefore, this region is also known as the "Third Pole" [2]. Water-filled pocket-like structures called cryoconites are present over the glacier's ablation zone [3,4]. They contain a dark-colored granular material, which consists of organic and inorganic matter [5–7]. Studies reporting on the diversity of polar and alpine cryoconite holes suggest the presence of vast microbial communities, including photosynthetic cyanobacteria and algae, heterotrophic bacteria, viruses, yeasts, diatoms and metazoa, most of which showed extremely good cold adaptation and growth characteristics [8–10].

In glacier cryoconite ecosystems, microbes play an important role [11,12], but the Himalayan glacier cryoconites and their ecosystem still represent a gap in investigations. In these natural ecosystems, microorganisms able to survive in low temperatures are widespread, where they often represent the dominant flora, and as a result, they should be regarded as the most flourishing colonizers of our planet [13]. These cold environments are mostly dominated by cold-adapted (psychrophilic) and cold-tolerant (Psychrotolerant) microbes. Cold-adapted microbes have common physiological and cellular features, such

as colony pigmentation, ability to grow at low temperatures with low nutrient concentrations, specific membrane structure, synthesis of cryoprotectants, exopolymers, cold-active enzymes and ice-binding proteins.

According to Margesin et al. [14] and Bej & Mojib [15], the extremely cold Himalayan glaciers were considered a barren habitat for microbial life. However, the current study has shown that there are many organisms that colonize in high-altitude Himalayan cryoconites, and they thus represent a hot spot for the assessment of cryophilic microbial biodiversity. Various distinct groups of microbes have been isolated and characterized from the soil and river habitats of the Himalayas [16–18]. However, studies on the existence of yeasts and molds in cryoconite holes of the glacier and their characterization remain lacking as yet.

The proposed work was focused on studying the fungal diversity in glacial cryoconite samples collected from the Hamtah glacier, Himalaya. Characterization was performed using a combined approach of physiological, biochemical and molecular analyses. These microhabitats are distinct and have not been thoroughly investigated before in order to discover the microbial diversity. Since true psychrophiles live in these environments, it was very desirable to concentrate the exploration on the fungal diversity and potential in these climatically extreme microhabitats.

In this study, the results regarding the isolation, identification, adaptation strategies and biotechnological potential of culturable yeasts and filamentous fungi from cryoconite holes of the Hamtah glacier are presented. The present study also focused on morphological, physiological and molecular parameters. D1/D2 and internal transcribed spacer (ITS) regions were investigated. The isolate has been further examined for enzyme screening to acknowledge adaptation strategies and biotechnological prospects (Supplementary Figure S1).

## 2. Materials and Methods

### 2.1. Cryoconite Sample Collection and Isolation of Fungi

The Hamtah glacier (32.24° N, 77.37° E) that was selected for analysis is a simple glacier with a single lobe, extending from south to north between 5000 and 4020 masl, covering an area of about ~3 km$^2$ and is ~6 km long [19] located in the Chandra Basin on the northern slopes of the Pir-Panjal range of the Himalayas, in the Lahaul-Spiti valley of Himachal Pradesh, India. The seven cryoconite hole samples were collected from different locations on the Hamtah glacier. The temperature and pH of the water in the cryoconite holes were measured (Supplementary Figure S2). Cryoconite samples were collected by aspirating spherical dark granular material (debris) using a sterile 50 milliliter syringe, placed in sterile ampules (Himedia). The sampling was performed following contamination-free procedures as suggested by Veysseyre et al. [20]. All the samples were stored in an insulated box, at sub-zero temperatures, till further analyses. For the isolation of fungi, one gram (wet weight) of cryoconite samples was processed in the laboratory following the serial dilution method [21], and plated on various agar plates viz. rose Bengal agar (RBA), potato dextrose agar (PDA) and malt extract agar (MEA) (Hi-Media) aseptically and incubated at 1, 4, 15 and 22 °C (low-temperature incubator, MIR 253, Sanyo, Japan). The emerging fungal colonies were isolated and purified by repeated streaking on fresh media. The purified isolates were then stained and observed for morphological details under an epifluorescence research microscope (BX51 Olympus, Japan). The morphological and biochemical features of the yeasts were also investigated following standard procedures [22]. The purified isolates were maintained on PDA slants at 4 °C and in Glycerol stock at −20 °C.

### 2.2. DNA Extraction, PCR, Sequencing, and Phylogenetic Analysis

Total DNA was extracted from the culture grown on the PDA plate for three weeks at 4 °C. Yeast cells were homogenized and their DNA was extracted with an ISOPLANTII kit (Wako pure chemical industries, LTD, Osaka, Japan). The extracted DNA was amplified by the PCR method using KOD-plus DNA polymerase (TOYOBO Co., Ltd., Osaka, Japan). The

final concentration of PCR reaction mixture was 10–50 ng (extracted DNA, 0.2 mM dNTPs, 2.0 mM MgSO$_4$, 0.3 μM primers and 1 U KOD-plus DNA polymerase). The ITS region was amplified using the primers ITS1F (5′-GTA ACA AGG TTT CCG T) and ITS4 (5′-TCC TCC GCT TAT TGA TAT GC) for the identification of filamentous fungi. However, the D1/D2 domain (NL1 (5′-GCA TAT CAA TAA GCG GAG GAA AAG) and NL4 (5′-GGT CCG TGT TTC AAG ACGG) was amplified for identification of yeasts. PCR thermal cycler conditions (5 min at 94 °C, followed by 30 cycles for 15 s at 94 °C, 30 s at 54 °C and 90 s at 68 °C) were followed. The PCR products were checked by electrophoresis using 1.5% (*w/v*) agarose gel. Sequences were obtained with an ABI prism 3100 Sequencer (Applied Biosystems) using the ABI standard protocol.

The ITS and D1/D2 sequences were aligned through MAFFT [23], analyzed in MEGA 7 [24], and Bayesian inference trees were constructed through MrBayes 3.2.5 [25]. A 50% majority-rule consensus tree was calculated to estimate the posterior probabilities. Tree nodes were tested via bootstrap analysis, and a bootstrap percentage ≥50% or Bayesian posterior probability ≥0.9 were considered supportive. The neighbor-joining method was used to construct phylogenetic trees. Gene sequences of yeast and filamentous fungal strains were submitted to NCBI GenBank, and the accession numbers assigned were used in phylogenetic trees.

### 2.3. Physiological Characterization

The yeast physiology study was performed by inoculating a loopful of yeast in a flask containing 50 mL of potato dextrose broth (PDB). The flask was incubated at 1, 4, 15, and 20 °C, at 150 rpm in a refrigerated shaking incubator (Refrigerated incubator shaker, IS-971RF, JeioTech, Daejeon, Korea). After 1 week of incubation, absorbances were taken at 600 nm. The growth of yeast isolates was tested in potato dextrose broth with varied pH values (3, 5, 7 and 9) and NaCl concentrations (0, 2.5, 4.5 and 8.5% *w/v*) at 15 °C and 150 rpm in a refrigerated shaking incubator (Refrigerated incubator shaker, IS-971RF, JeioTech, Daejeon, Korea). The growth was analyzed spectrophotometrically at 600 nm, the OD was converted to cell numbers and the CFU count was calculated. Filamentous fungal isolates (mold) were also tested for growth at different temperatures (1, 4,15 and 20 °C), pH values (3, 5, 7 and 9), and NaCl concentrations (0, 2.5, 4.5 and 8.5% *w/v*) on potato dextrose agar at 15 °C, and the results were recorded regarding growth after 1 week of incubation.

### 2.4. Carbohydrate Assimilation Test

Yeast isolates from the culture broth were inoculated in a Hi-Carbohydrate kit (KB009, Hi-media) for carbon source utilization tests, and the results were noted as per the instructions. For filamentous fungal isolates oxidation fermentation, basal medium was used for the assimilation of carbohydrates with the incorporation of different sugars.

### 2.5. Antibiotic Susceptibility Test

The antibiotic susceptibility of the isolates was tested by the disc method [26]. Fungal cultures were grown in 20 mL of potato dextrose broth for 1 week at 15 °C. In total, 0.2 mL of the culture suspension was spread-plated on Muller Hinton agar. A paper disc containing various antibiotics, such as amphotericin B (AP-20), clotrimazole (CC-10), fluconazole (FLC-10), itraconazole (IT-10), ketoconazole (KT-10), miconazole (MIC-30) and nystatin (NS-50), was placed over the spread fungal inoculum and the plates were kept for incubation at 15 °C for 1 week. The zone of inhibition (in millimeters) was detected and recorded.

### 2.6. Screening of Enzymes

Substrate-specific media was used for the screening of enzymes by the fungal isolates. The pH of all media was maintained at 5.5. For cellulase screening, 1% carboxy methylcellulose powder was added to the mineral salt solution [27,28]. The production of extracellular protease was determined using 1% skimmed milk. The amylase test was done using 1% starch added to the mineral salt solution. The lipase test was done using

Tween 80 as the substrate. All the isolates were spot-inoculated on their respective agar plates and each isolate was incubated at various temperatures of 1, 5, 10, 15, 20 and 25 °C. Enzymatic hydrolysis by the fungal isolates was indicated by the zones of clearance around the colonies, which were measured in mm as the difference between the diameter of the halo zone and the colony. Catalase tests were also performed for the yeast isolates, and their activity was determined by the addition of 3% hydrogen peroxide to the loopful of organism on the microscopic slide. Positive catalase reactions are indicated by immediate effervescence (bubble formation).

## 3. Results

### 3.1. Isolation of Culturable Fungi

For the basic understanding of fungal biodiversity, an attempt was made to isolate and characterize different yeasts and filamentous fungi associated with the cryoconite holes of the Hamtah glacier, Himalaya. The present study revealed that the temperatures and pH values of the cryoconite holes showed variations. The temperatures varied from 1.2 to 2.1 °C and the pH ranged from 7.7 to 8.5. A diverse group of psychrophilic and psychrotolerant fungi were isolated and purified from different cryoconite samples (A, B, C, D, E, F and G) from the Hamtah glacier, Himalaya. In total, 45 yeast isolates and 10 filamentous fungal isolates were isolated. The morphology of the yeast cells and the morphotaxonomic characteristics of the filamentous fungi were studied using light microscopy, and a few of them are depicted in Figure 1.

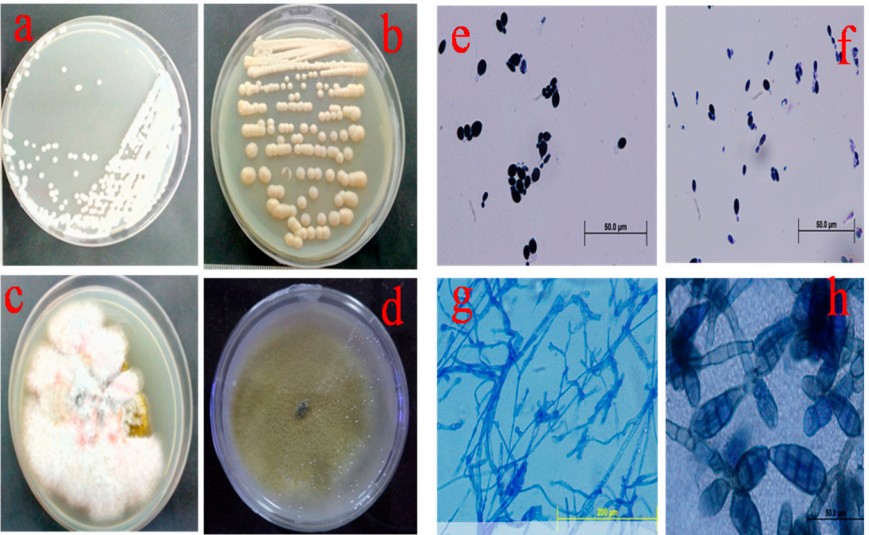

**Figure 1.** Yeast colonies (**a**,**b**), filamentous fungal colonies (**c**,**d**) grown on culture media. Micro morphology of yeast (**e**,**f**), filamentous fungi (**g**,**h**) observed under 100×.

### 3.2. Phylogenetic Analysis

In total, the 11 fungal genera identified were *Mrakia, Goffeauzyma, Glaciozyma, Cladosporium, Alternaria, Penicillium, Yarrowia, Coniochaeta, Lecanicillium, Rhodotorulla* and *Phenoliferia* (Figure 2a–e). Yeast strains belonging to the genus *Mrakia, Goffeauzyma* and *Glaciozyma* were more dominant than the filamentous fungi, and varied according to their different locations.

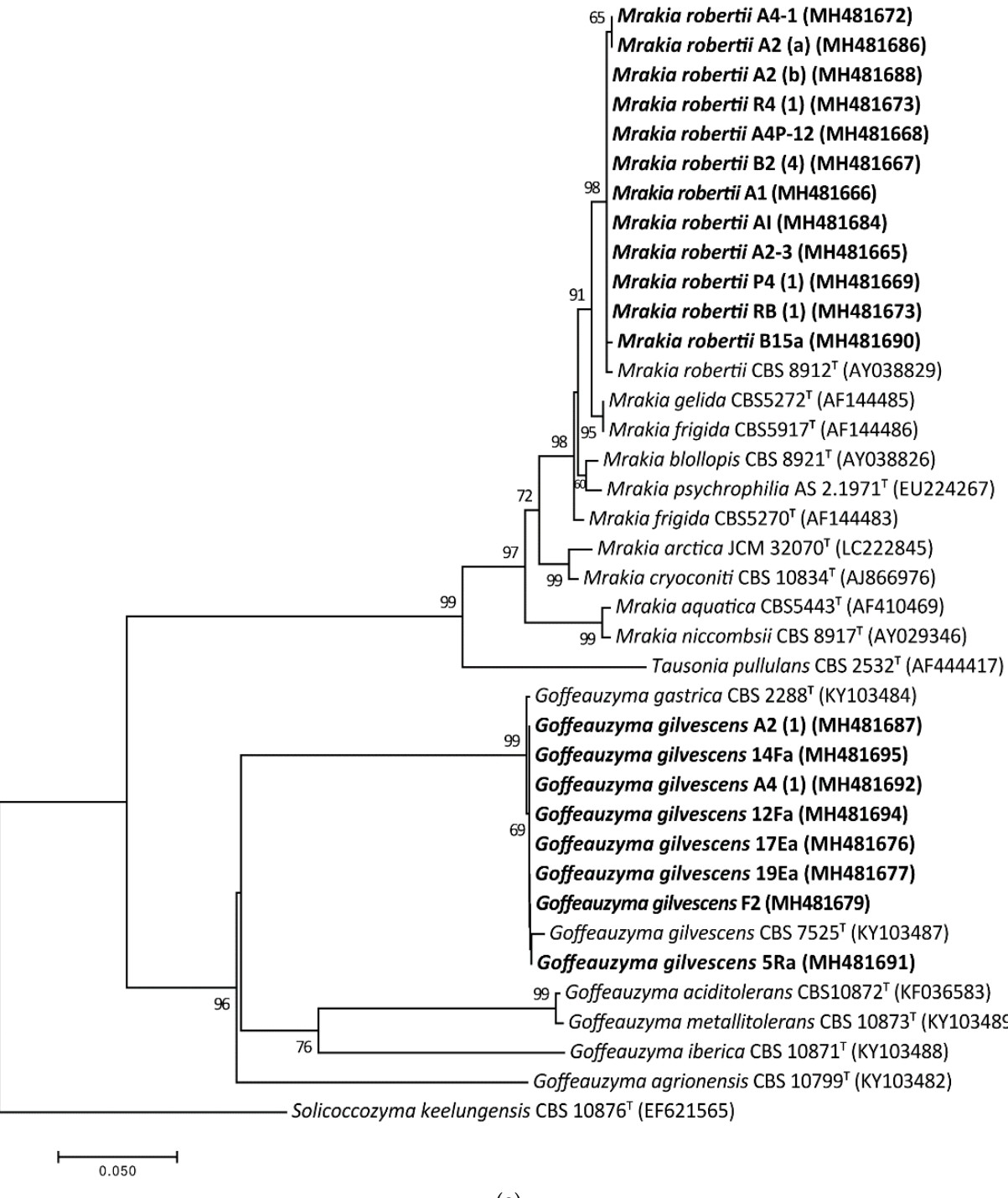

(**a**)

**Figure 2.** *Cont.*

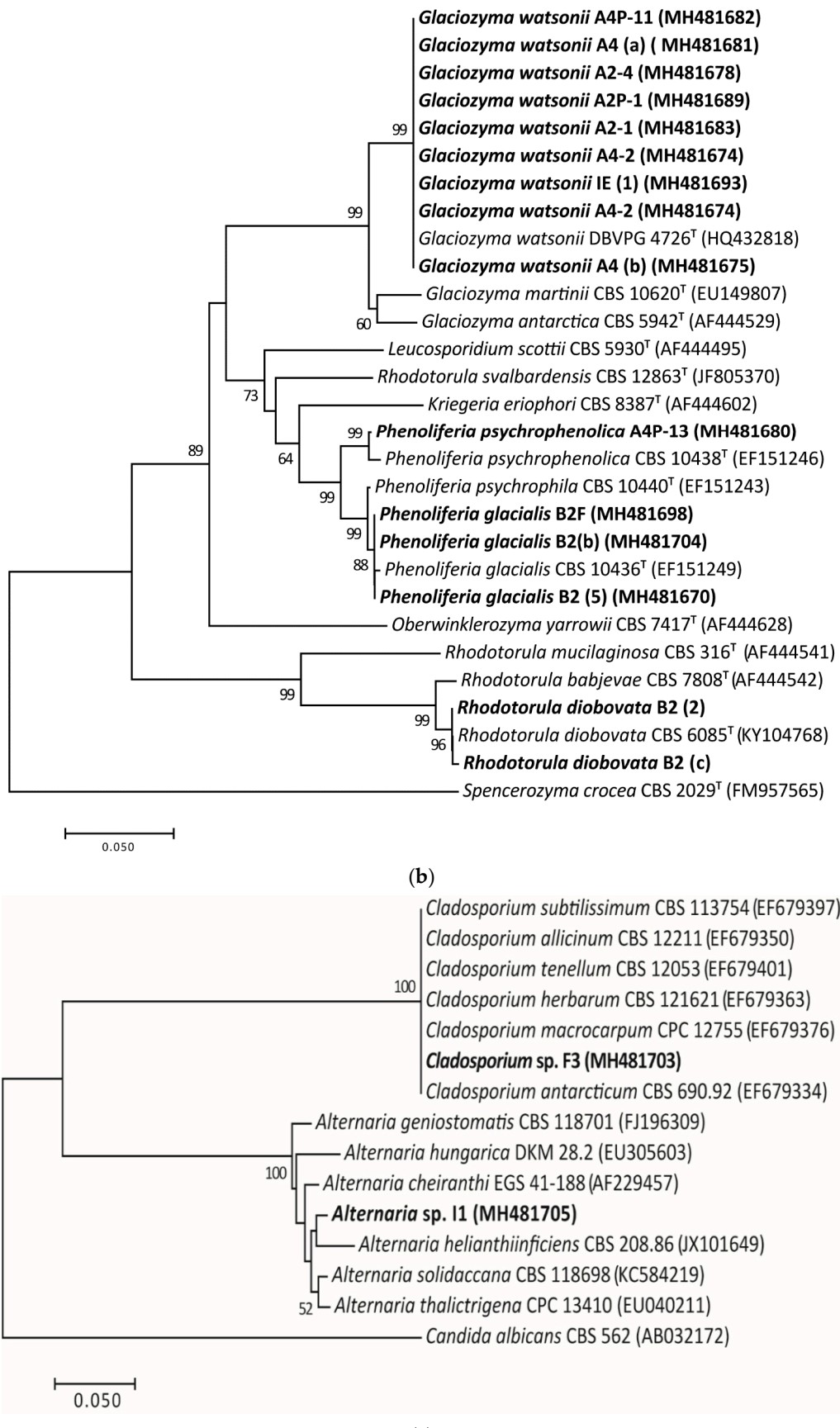

(**b**)

(**c**)

**Figure 2.** *Cont.*

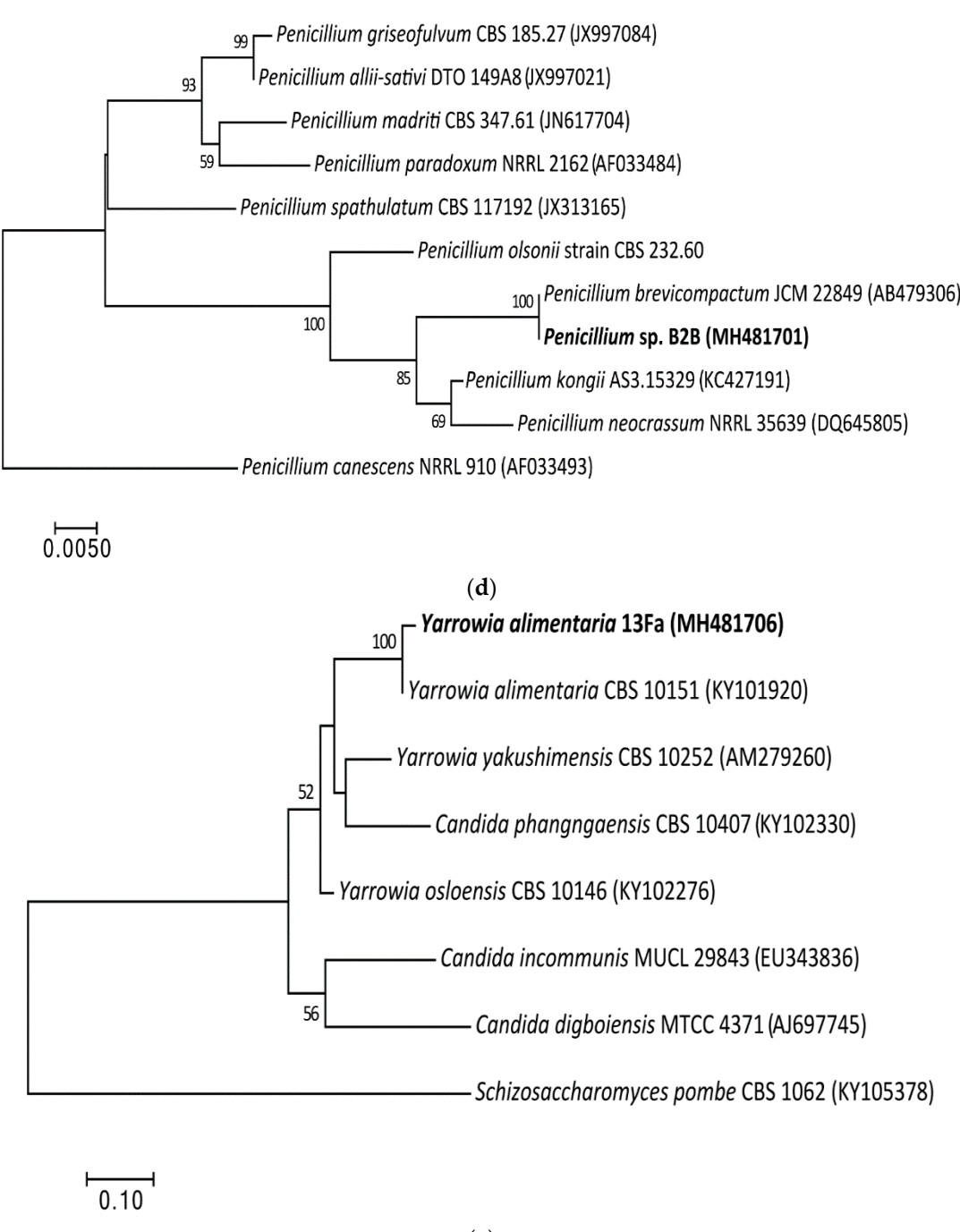

(**d**)

(**e**)

**Figure 2.** (**a**) Phylogenetic tree constructed showing the relationship between *Mrakia* sp. and *Goffeauzyma* sp. yeast strains from the Himalayas. (**b**) Phylogenetic tree constructed showing the relationship between *Glaciozyma* sp. and *Phenoliferia* sp. (**c**) Phylogenetic tree constructed showing similarity between *Cladosporium* sp. and *Alternaria* sp.; *Candida Albicans* was used as an out-group. (**d**) Phylogenetic tree constructed showing the relationships of *Penicillium* sp.; *Penicillium canescens* was used as an out-group. (**e**) Phylogenetic tree constructed showing the relationships of *Yarrowia alimentaria*. *Schizosaccharomyces pombe* was used as an out-group.

### 3.3. Growth Temperature, pH, and Salt Tolerance of Fungal Isolates

Most of the isolates showed growth temperatures ranging from 1 to 20 °C, with the optimum temperature for growth at 15 °C, which indicates the psychrotolerant nature of the isolates. The isolates showed low growth at pH 3.0, and comparatively, the maximum growth was achieved between pH 5.0 and 7.0. The salt tolerance test showed visible growth

in non-saline media, 2.5% and 4.5% *w/v* NaCl, but beyond that, at 8.5%, there was a fall in growth (Supplementary Table S2).

### 3.4. Carbohydrate Utilization of Fungal Isolates

Out of the 35 carbon sources tested for yeast isolates, *Mrakia* strains and *Glaciozyma* sp. utilized the maximum number of carbon sources. Of all the yeast isolates screened, *Mrakia robertii* (A4-1 (MH481672) and *Glaciozyma watsonii* (A4P-11 (MH481682) and A2-1 (MH481683) utilized 25 carbon sources (Figure 3a,b), including lactose, xylose, maltose, dextrose, galactose, glucoside, mannose, mannitol, sorbitol, esculin, sorbose, arabinose, citrate, malonate, cellobiose, adonitol, rhamnose and arabitol. Filamentous fungi also showed varied responses in terms of their ability to utilize various carbon sources. Filamentous fungal strains of *Coniochaeta* sp. were able to utilize the maximum number of carbon sources tested (Figure 3c), and were found positive for glucose, mannose, lactose, rhamnose, cellobiose, dextrose, galactose, arabitol, adonitol, sorbose, arabinose, inositol, raffinose and dulcitol.

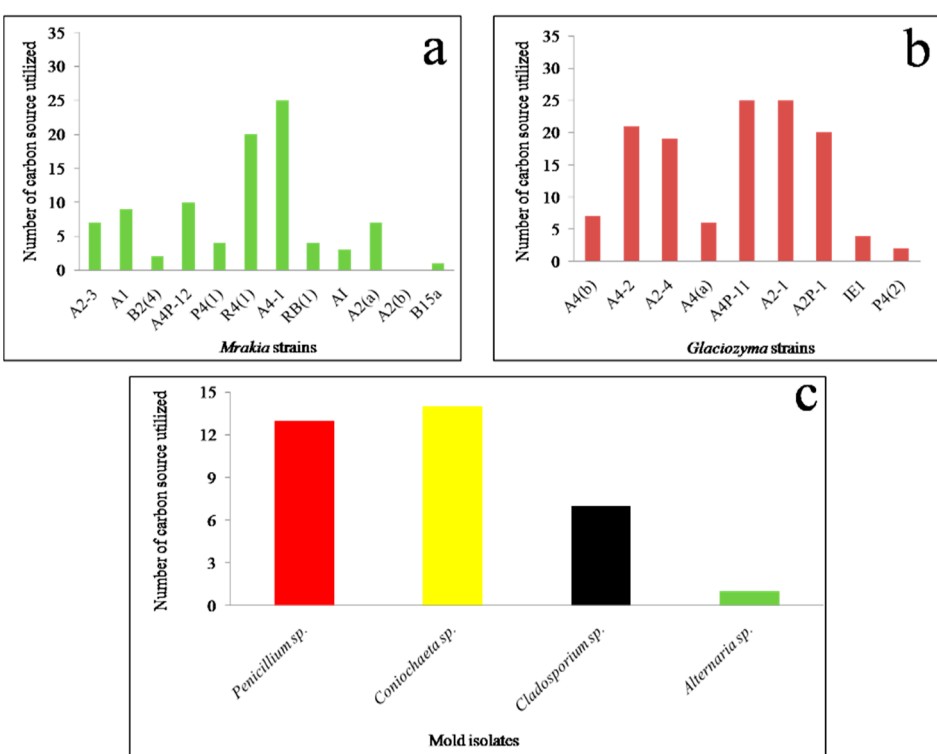

**Figure 3.** Cryoconite fungal isolates showing the capacity for carbon source utilization. (**a**,**b**) Yeast isolates from *Mrakia* and *Glaciozyma* strains showing capacity for carbon source utilization. (**c**) Mold isolates showing capacity for carbon source utilization.

### 3.5. Antibiotic Resistance and Sensitivity Patterns of the Isolated Strains

All 55 fungal isolates were screened for antibiotic sensitivity and resistance patterns. The results show that most of the isolates were sensitive towards the seven antibiotics tested. Most of the fungal cultures tested were found susceptible to Miconazole and Amphotericin B (Figure 4a). Among the yeast isolates, *Glaciozyma* sp. (A2-1), *Rhodotorula* sp. B2(d), *Rhodotorula* sp. B2(2) and *Goffeauzyma* sp. (12Fa) were found to be sensitive to all seven antibiotics tested. *Yarrowia* sp. (13Fa) proved to be the most resistant, showing susceptibility to one (clotriconazole) out of the seven antibiotics screened. The filamentous fungi *Alternaria* sp. I1 and *Penicillium* sp. B2B were the most susceptible, while other isolates showed varying resistance patterns (Figure 4b) (Supplementary Table S4).

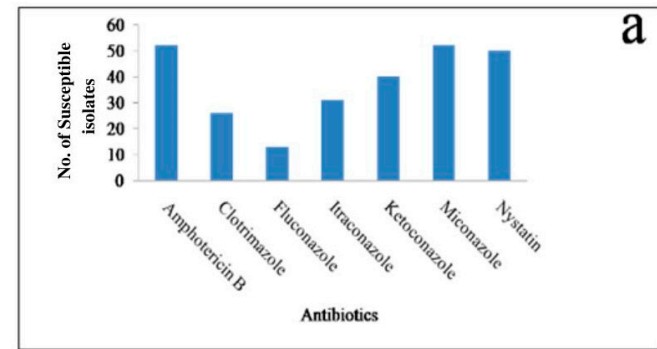

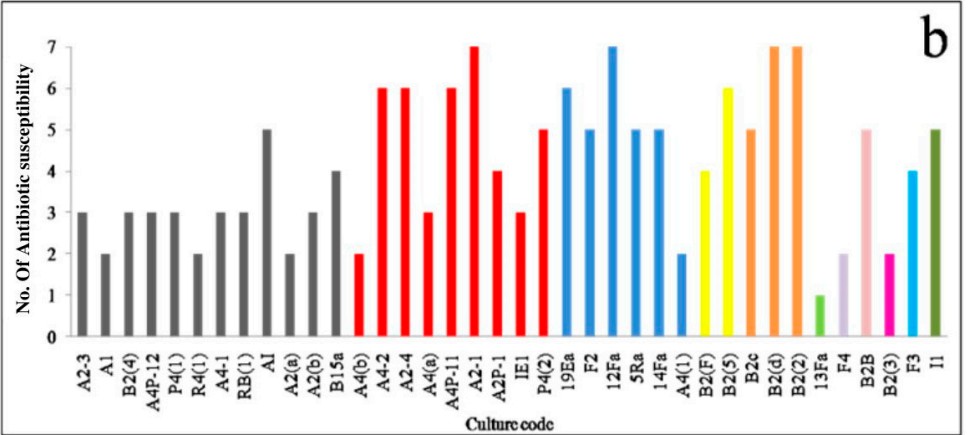

**Figure 4.** Antibiotic resistance pattern of cryoconite fungal isolates. (**a**) Number of isolates showing sensitivity to seven tested antibiotics. (**b**) Sensitivity pattern of each isolate towards seven tested antibiotics.

### 3.6. Extracellular Enzyme Activities

The yeast and fungal isolates screened for their enzymatic activities (amylase, cellulase, lipase, protease) are depicted in Figure 5a–d. Most of the fungal isolates analyzed showed the maximum cellulase activity. Higher amylase, cellulase, and protease activities were observed in *Mrakia* sp. Lipase activity was observed at its highest in *Goffeauzyma* sp. *Mrakia* sp. A2-3 and *Yarrowia* sp. 13Fa were the best cellulase producers among the yeast isolates. All the isolates were found positive under the catalase test when tested with hydrogen peroxide.

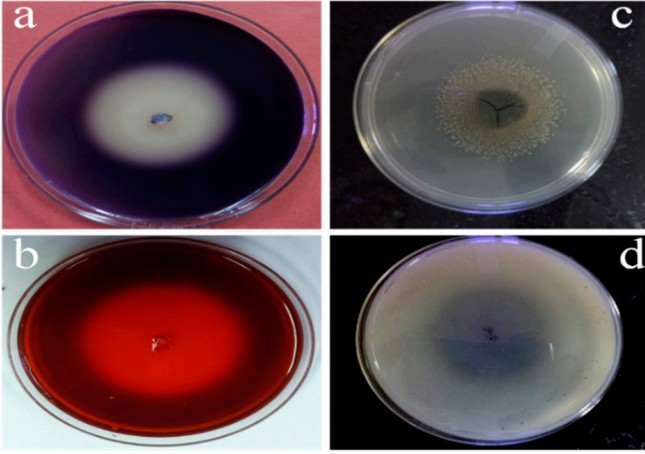

**Figure 5.** Plate assay screening of various enzymes. (**a**) Amylase activity; (**b**) cellulase activity; (**c**) lipase activity; (**d**) protease activity.

## 4. Discussion

The present work, which describes the fungal diversity and characterization of yeast and filamentous fungi from the Hamtah glacier, is likely the first report from the Himalayas, as little is known about the microbial communities of Himalayan cryoconite. Because each cryoconite hole has a unique temperature and pH value, the temperatures of the cryoconite holes in this investigation displayed differences that were remarkable. According to data from Arctic cryoconites [7], the results of the current isolation, investigation demonstrate that cryoconite holes support a rich assemblage of yeast species more often than filamentous fungi. Eleven genera of yeast and fungi were found in the Himalayan glacier's cryoconite as a result of the current investigation. In yeast isolates, *Mrakia* and *Glaciozyma* were the two most predominant genera. Two genera—*Mrakia* and *Rhodotorula*—have previously been reported from glacier sediments in the Arctic and Alpine regions [29].

To distinguish the strains, physiological traits and tests of carbohydrate assimilation were performed. Growth temperatures lower than 20 °C for *Mrakia* sp., *Phenoliferia* sp., *Glaciozyma* sp., *Goffeauzyma* sp., *Rhodotorula* sp., and *Yarrowia* sp. indicate that the isolates are naturally psychrotolerant and can endure cold temperatures. This is comparable to the research performed by Hall et al. [30]. The optimum temperature for the majority of the yeast isolates was determined to be 15 °C. Additionally, the *Mrakia* sp. isolated from soil samples on King George Island in the sub-Antarctic region was described in a report that was related to this [31].

In the oligotrophic environment of the cryoconite holes, carbon sources are a crucial prerequisite for microbial metabolism, which is constrained. This finding is specifically significant because ice is a very scarce source of carbon. The present study revealed that the majority of the fungal isolates tested could use simpler types of carbohydrates such as lactose, dextrose, xylose, mannose, and galactose as carbon sources, which is similar to the results of an earlier study from the Arctic [32]. This is true even though the microbial communities seen in glacier cryoconite samples were clearly different. In pristine conditions, Gardner et al. [33], observed the presence of antibiotic-resistant plasmids. Antibiotic resistance research in natural ecosystems, however, is still in its early stages. In the current investigation, it was shown that isolates' sensitivity to antibiotics differed. Similar findings were also made in the Arctic, where the strains have quite different properties related to their antibiotic sensitivity [11]. According to Martinez [34], the genes for antibiotic resistance have likely been present for billions of years, and are a form of adaptation that allows microbial strains to withstand adverse conditions and endure in the Arctic environment [11]. Since these substances make up the snowpack DOM (dissolved organic matter) pool, the capacity of the isolated strains to break down organic substances including cellulose, lipids, proteins, and carbohydrates is important [35]. According to the literature, little is known about the extracellular enzyme synthesis of Himalayan microbial strains. Since industrial fermentation occurs at room temperature, isolates were examined for a wide variety of cold extracellular enzymes. Psychrotolerants are therefore chosen over psychrophiles and mesophiles for the synthesis of enzymes. Overall, the findings demonstrate that each strain had a strong potential to metabolize the substrates found in the snowpack. This is in line with the theory that organic molecules in snow and ice serve as a source of carbon and energy for resident microbes [36,37]. According to Bjelic et al. [38], cold-active enzymes have reduced thermal stability that prevents freezing, and higher enzymatic activity at lower temperatures than their mesophilic equivalents [39]. The results of the screening revealed the cellulolytic nature of the isolates, and their optimum cellulolytic activity at 15 °C. This discovery is consistent with findings of Holding [40]. To utilize and recycle the significant organic carbon stock trapped in the soils of colder regions, strains with a cellulose-degrading activity at low temperatures can be extremely significant. The prevalence of substrate-degrading fungi suggests that the Himalayan cryoconite samples include significant amounts of protein, lipids, starch, and cellulose material, as well as the existence of fungal communities even under snow. This might be a particular adaptation to snowy environments, where the availability of organic substrates

is minimal. In turn, it shows that they are capable of actively influencing the chemistry of the snow through the mineralization of supraglacial organic matters.

## 5. Conclusions

The aim of the present study was to address a knowledge gap regarding the microbial communities found in Himalayan glacier cryoconites. The following genera of yeast and filamentous moulds were identified: *Mrakia*, *Glaciozyma*, *Phenoliferia*, *Goffeauzyma*, *Penicillium*, *Coniochaeta*, *Alimentaria*, *Yarrowia* and *Rhodotorula*. These diverse groups of Himalayan fungi have the potential to produce cold-active enzymes, which have considerable biotechnological significance.

**Supplementary Materials:** The following supporting information can be downloaded at: https://www.mdpi.com/article/10.3390/su142214814/s1, Figure S1: Schematic representation of research work; Figure S2: Exposed cryoconite holes over the Hamtah glacier Himalaya; Table S1: Sampling sites and properties of cryoconite sediments; Table S2: Physiological characteristics of the cryoconite fungal isolates; Table S3: Carbon source utilization of the cryoconite isolates; Table S4: Screening for antibiotic sensitivity of Cryoconite fungal isolates.

**Author Contributions:** Conceptualization, sampling, methodology, funding acquisition, supervision S.M.S.; formal analysis G.M.D. and M.T., writing—original draft preparation, G.M.D.; writing—review and editing, S.M.S. All authors have read and agreed to the published version of the manuscript.

**Funding:** This research received funding support from NCPOR, India and NIPR Japan.

**Institutional Review Board Statement:** Not applicable.

**Informed Consent Statement:** Not applicable.

**Data Availability Statement:** Not applicable.

**Acknowledgments:** The authors are grateful to the Directors, NCPOR, Goa and BHU for their facilities. The authors are also thankful to NCPOR-MoES and NIPR for the financial support.

**Conflicts of Interest:** The authors declare no conflict of interest.

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
