# Peer review of "Identification of Fungal Communities Isolated from Himalayan Glacier Cryoconites"

_sustainability, doi:10.3390/su142214814_

Round 1

Reviewer 1 Report

This is an interesting manuscript about the isolation, identification, and characterization of fungal communities of cryoconite holes of Hamtah glacier, Himalaya. Several species of yeast and filamentous molds were identified which belong to different genera by ITS and D1/D2 region sequences. Also, physiological characterization, carbohydrate assimilation test, antibiotic susceptibility test (e.g., amphotericin B, clotrimazole, fluconazole, itraconazole, ketoconazole, miconazole, and nystatin), and enzyme production (e.g., cellulase, lipase, protease, amylase, and catalase) were evaluated for the identified isolates. The manuscript shows novel results, and the objectives are very clear.

Reviewer 2 Report

The manuscript entitled Characterization of Fungal communities of Himalayan glacier Cryoconites presents data related to the isolation of yeast and fungal strains from Himalayan glaciers. The information presented is not well organized, the methods need to be better depicted and the results better presented. Also, there are several English tupo and grammas mistakes that the authors must revise. Below are the comments.

-The title must be Identification of fungal communities isolated from Himalayan glacier cryoconites.

Line 30. The statements must start with letters, not with numbers. Revise this mistake along the manuscript.

Line 67. How many cryoconites were considered? In the abstract, it was mentioned 10 samples but in the materials and methods section, it was mentioned 7 samples.

Line 67. Variouslocations? Separate the words.

Figure 1 is not well organized. What is the rationale for showing figure 1? The information must be mentioned in the materials and methods section.

Line 74. Separate the values from units. Revise the same situation in all the manuscript.

Section 2.2. Improve the depiction of the methods. What was the length of the sequences? What was the ITS gene used? What was the rationale for using both ITS and D1/D2?

Section 2.6. Depict in dept the procedures for the evaluation of the enzyme production.

Figures. Revise the correct form of presenting the caption for the figures. Revise the instructions for authors.

Figure 2. The organization and the depiction of the figure are not clear.

Figures 3a-3e. Must be better organized and depicted. Why the strains were not identified to the species level? Since authors amplified ITS and D1/D2 regions?

Figure 4. The information shown in the figure is not clear. Give more details of the figure in section 3.4. What are the abbreviations A2-3, A1, B2(4), etc.?  I recommend analyzing the data by Principal Component Analysis in order to show the data in a single graph. 

Figure 5. Also, the data must be analyzed by PCA.

Section 3.6. What was the rationale for assaying the enzymatic activity of the isolated strains? The section must be deleted since no relevant data were obtained.

Author Response

Please find attached file for our response to reviewer comments.

Reviewer 3 Report

Authors reported in the present manuscript characterization of Fungal communities of Himalayan glacier 2 Cryoconites.

There are the following suggestions as follows:

1. In the last introduction section, it is recommended to include the schematic rationale of the research. 

2. To the improvement of quality of morphology, authors must include the SEM image of fungi after incubation for 24h.

3 Discussion part is needed for more improvement.

4. The background of each image in Fig 6 must be the same. 

Author Response

Please find attached response file for reviewer comments

Round 2

Reviewer 2 Report

The manuscript has been improved. However, there are issues that authors must attend to or clarify.

-The authors included two figures in lines 86 and 127 that must be included as supplementary material.

-Also, include Figure 1 as supplementary material since there is no depiction or relevance for showing it in the main text. 

-Authors must improve the phylogenetic analysis in order to identify the strains to specie level.

-Authors rebutted that the data shown in figures 4 and 5 cannot be analyzed by PCA. However, the way they show the data is not clear to understand by the readers. Authors must find other ways to analyze or show the data.

Author Response

Please find attached responses to reviewer comments

Round 3

Reviewer 2 Report

The authors attended to almost all the comments. However, the authors MUST re-run the phylogenetic analysis or interpret the results they are presenting. They have strains such as Penicillium sp. B2B (MH481701) is located in the same clade as Penicillium brevicompactum. Why they can not be considered the same species? The same comment is for Alternaria sp. I1 (MH481705) and Cladosporium sp. F3 (MH 481703).  The authors rebutted they will examine the strains in the future. Hence, they must delete the information from the manuscript and report those strains in the future.   

Author Response

In the present MS we are contributing fungal taxa present in Himalayan glacier cryoconites and their potentials in extreme environment. Therefore, finer taxonomic analyses of all the strains were not performed for all the genera.